# A Comprehensive Review on Processing, Development and Applications of Organofunctional Silanes and Silane-Based Hyperbranched Polymers

**DOI:** 10.3390/polym15112517

**Published:** 2023-05-30

**Authors:** Balaraman Indumathy, Ponnan Sathiyanathan, Gajula Prasad, Mohammad Shamim Reza, Arun Anand Prabu, Hongdoo Kim

**Affiliations:** 1Department of Chemistry, School of Advanced Sciences, Vellore Institute of Technology, Vellore 632 014, Tamil Nadu, India; indumathy.b2019@vitstudent.ac.in; 2Department of Advanced Materials Engineering for Information & Electronics, College of Engineering, Kyung Hee University, Yongin-si 17104, Republic of Korea; 3School of Energy, Materials and Chemical Engineering, Korea University of Technology and Education, 1600, Cheonan-si 31253, Republic of Korea

**Keywords:** silanes, hyperbranched polymer, synthetic strategies, applications

## Abstract

Since the last decade, hyperbranched polymers (HBPs) have gained wider theoretical interest and practical applications in sensor technology due to their ease of synthesis, highly branched structure but dimensions within nanoscale, a larger number of modified terminal groups and lowering of viscosity in polymer blends even at higher HBP concentrations. Many researchers have reported the synthesis of HBPs using different organic-based core-shell moieties. Interestingly, silanes, as organic-inorganic hybrid modifiers of HBP, are of great interest as they resulted in a tremendous improvement in HBP properties like increasing thermal, mechanical and electrical properties compared to that of organic-only moieties. This review focuses on the research progress in organofunctional silanes, silane-based HBPs and their applications since the last decade. The effect of silane type, its bi-functional nature, its influence on the final HBP structure and the resultant properties are covered in detail. Methods to enhance the HBP properties and challenges that need to be overcome in the near future are also discussed.

## 1. Introduction

Hyperbranched polymers (HBPs) are a type of macromolecule that has gained greater attention recently because of their unique framework and attributes [1,2,3,4,5,6,7]. HBPs are defined by a highly branched architecture, with many short branches radiating from a central core [8,9,10,11]. As HBPs are characterized by a complex, branching structure that resembles a dendritic tree and the presence of enormous end groups, HBPs exhibit several advantages over conventional linear polymers, including low viscosity, improved solubility, high surface area, low polydispersity and tunable mechanical properties [12,13,14,15]. 

The ability of HBPs to be functionalized by contributing reactive groups at the end of their branches is one of their essential vital aspects. This allows for the covalent attachment of active compounds or other materials by further tailoring their properties for specific applications. These properties make HBPs a promising class of materials with potential applications in a wide variety of fields, including drug delivery systems, surfactants, precursors, sensing, catalysis and material science [16,17,18,19,20]. 

HBPs exist as an irregular branching structure, provided with unique properties and a three-dimensional (3D) geometrical structure with a degree of branching (D.B.) less than 1.0, low molecular chain ambiguity, increased solvability, lower viscosity, intramolecular cavities and large functional end groups, which is accountable for its use in several applications including supramolecular chemistry, bio-medicinal drug delivery, surface coatings, additives, crosslinkers, optical, electrical, sensors and magnetic materials [21,22,23,24,25,26,27,28,29,30]. The presence of intramolecular cavities in their 3D globular architecture can be utilized to encapsulate medicines, and due to their ease of synthesis in comparison to dendrimers, HBPs have attracted attention for the development of drug delivery systems. 

On the other hand, silanes are a class of hybrid organic-inorganic compounds. The basic components of silanes are monomeric silicon (Si) compounds with four substituent groups attached to the Si atom, which can be any combination of reactive or non-reactive inorganic or organic groups, which are the fundamental building blocks of silanes. R_n_Si(OX)_4-n_ is the basic framework of an organosilane, in which “R” denotes an organo-functional group and “OX” denotes an alkoxy or acetoxy group. Regarding their hydrolyzable substituents, the majority of silanes are typically tri-functional (R-Si-X_3_); X is typically an alkoxy substituent such as -OCH_3_ and -OCH_2_CH_3_. However, the Si functional group can either have two or just one “X” substituent. The mono-alkoxysilane can only create a single layer, whereas the di-alkoxy and tri-alkoxy silanes can produce multi-layered interphases after being hydrolyzed to their silanol forms [31,32]. 

Due to their potential applications, Si-containing HBPs (Si-HBPs) have generated much attention among researchers. In silanes, when two or more reactive groups of different kinds are held with the Si atom in a molecule, the generated compound is called organofunctional silane. One of the reactive groups (alkoxy or hydroxy) in organosilanes reacts with different kinds of inorganic materials such as glass, metals, silica and sand as a result of its chemical bonding with the surface of inorganic material. The other reactive group, such as vinyl, epoxy, methacryl, amino and mercapto groups, react with different kinds of organic materials or synthetic resins through a chemical interface. These two different kinds of reactive groups in silanes have made them capable of forming chemical bonds with both inorganic and organic materials, which is responsible for their dual functionality [33,34]. 

Organofunctional silanes can fulfill the minimum requirement to produce an HBP by readily reacting with the monomer due to its hydrolyzable nature with most organic/polymeric materials. Developments of such silane-based HBPs are required to address the complications arising from the usage of conventional core molecules. Silane HBPs can be effectively used as surface treatment material and as an excellent adhesive, and they can also be used directly in the polymeric material due to their ability to bond within the polymer matrix.

The purpose of this review article is to provide a general outline of the latest developments in HBPs, particularly in Si-HBPs. In recent years, many researchers have been interested in the development of HBPs. Silanes are favorable for the improvement of the properties of HBPs due to their dual reactivity nature. Therefore, this review focuses on the effects of different types of silanes on the improvement of the properties of HBPs, which are mainly attributed to the type of silane used in the HBP synthesis. Meanwhile, some relationships among the functional groups present in silanes are also discussed. Eventually, this review aims to give detailed information on choosing an appropriate silane type for the synthesis of HBPs with enhanced properties. 

Studies on the topics of “silane HBPs” and “HBPs” were explored on the Web of Science portal, and the outcomes are displayed in Figure 1. Every year, more and more articles pertaining to the topic “HBPs” are published on the platform. From 2012 to 2022, a total of around 500 publications per year were retrieved on the topic of “HBPs”, which demonstrates the topic’s high relevance. On the other hand, the number of articles that addresses the topic of “Si HBP” is not more than five articles per year in the portal. This result illustrates a wider space for improvement of Si-based HBP synthesis owing to its application in diverse fields.

## 2. Overview of Silanes

In the Periodic Table, Si belongs to major group 14 along with other elements such as Carbon, Germanium, Tin, Lead and Flerovium, and it has four valence electrons. Si differs chemically from the other elements of Group 14 in terms of structure, reactivity, and, therefore, physical and chemical properties due to its vacant 3D orbitals. Si is classified as a metalloid, as some of its properties are similar to both metals and nonmetals. In the Earth, Si can be found as silica, a variety of silicates and aluminous silicates. With other carbon atoms, carbon can form indefinitely long chains as (-C-C-)_n_-. On the other hand, Si cannot form long chains but can bond to other Si atoms to form shorter chain lengths, which are unstable in nature [35,36]. 

This is because the bond energy of the C-C bond is 356 kJ/mol, which is substantially greater than the bond energy of the Si-Si bond (226 kJ/mol). Yet, Si can create arbitrarily long chains to form a siloxane linkage (-O-Si-O-) when combined with oxygen atoms due to the extremely high Si-O bond energy (286 kJ/mol). Si compounds are more highly reactive than carbon compounds as a result of the vacant 3d orbital. 

Silanes are compounds made of monomeric Si. An organosilane is a silane with at least one Si-C link (Si-CH_3_). Through a sequence of reactions, organofunctional silanes are created from SiO_2_ (silica), the most prominent mineral on our planet Earth. During the reduction of silica to Si, trichlorosilane (HSiCl_3_) is produced through a reaction between Si and HCl. To create the functional silanes, trichlorosilane first combines with an alkene and then undergoes alcoholysis or a reaction with alcohol. 

### 2.1. Chemistry of Silanes

Trialkoxysilane, also termed a silane coupling agent, has three alkoxy functional groups at the extremities of its molecular backbone that bind an inorganic substrate to an unpolymerized resin matrix (surface). “R” is an organofunctional group such as amino, vinyl, aryl, methacrylate, acrylate, isocyanato, or sulfo, which furnishes the organic affinity that enables silane groups to produce interpenetrating networks (IPNs). R-(CH_2_)_n_-Si-(OX)_3_ is the general formula of these bi-functional, organofunctional silanes, where “R” is an organofunctional group [37]. 

Organofunctional silanes are molecules with two distinctly reactive groups linked to the Si atom as shown in Figure 2. This allows them to react and couple to an inorganic surface (such as ceramics and oxide coatings on metals) or to organic resins via a covalent bond. Organofunctional trialkoxysilanes often have the following molecular structure:R′(CH_2_)_n_Si(OX)_3_, where n = 0, 1, 2, 3 …. (1)

The above Si molecule contains two important varieties of reactive groups: (1) an organofunctional group or organic group (R′) like epoxy, amino, methacryloxy or sulfide, and (2) a hydrolyzable or leaving alkoxy group (OX) like methoxy (-OCH_3_), ethoxy (-OC_2_H_5_), and acetoxy (-OCOCH_3_). These functional and leaving groups are responsible for the organic compatibility that enables the silane to form IPNs in polymers. -(CH_2_)- acts as a linker (spacer) group that sits between the organofunctional groups R’ and the Si atom, and the hydrolyzable alkoxyl group is denoted as OX (methoxy, ethoxy). 

Before they may connect to the inorganic substrate, silanes are activated by an acid (acetic acid) or undergo hydrolysis to generate silanol groups (SiOH). Two types of moieties are bonded to the Si atom in an organofunctional silane. Alkyl silanes and aryl silanes are used to enhance gloss, concealing ability, mixing time and other attributes linked to their enhanced dispersion. They are additionally used to create hydrophobic surfaces for applications such as water repellents. The “X” stands for alkoxy molecules, most frequently methoxy or ethoxy, which interact with different hydroxyl groups to release methanol or ethanol. To increase coating integrity and adherence, these organofunctional groups (Table 1) offer the linkage with substrates, pigments, matrixes or fillers. The hydroxy functional polymers can also react with the methoxy groups [39,40,41,42,43,44,45]. These organofunctional trialk4oxysilanes undergo two important reactions such as condensation and hydrolysis [41].

Hydrolysis: hydrolyzable groups (-OCH_3_ or –OC_2_H_5_) undergo hydrolysis in the presence of water to form silanols (Si-OH). 

Condensation: silanols (Si-OH) are condensed together to form a siloxane structure (Si-O-Si), as shown in Figure 3 [41].

Meanwhile, the bonding of organofunctional silane molecules to polymers has been studied. Two mechanisms are thought to be responsible for silane’s effective adhesion to polymers: (1) chemical interactions between the reactive groups in the polymers and the organofunctional groups and (2) emergence of IPNs at the silane/polymer interface.

In general, organofunctional silanes act like a “bridge” to encourage adherence between polymers and inorganic substrates (such as glass or oxide coatings on metals). Functional silanes react with polymers to produce chemical bonds and IPNs for good silane/polymer adhesion. In addition, they react with inorganic surfaces to form metallo-siloxane covalent bonds for high adhesion between silanes and inorganic substrates. By reducing water infiltration and bond displacements at the fiber/resin interface, some organofunctional trialkoxysilanes were observed to greatly increase the mechanical strength of composite materials. Organofunctional silanes are now widely used in coatings/paints, adhesives and sealants [43,44].

Before bonding to the substrate, these functional silanes must first be activated by hydrolysis to form silanols. The initial step in the hydrolysis of silanes to silanols (SiOH) is the protonation of alkoxy groups present in silanes. The central Si atom is then the site of a bimolecular nucleophilic substitution (SN_2_) process. First, a water molecule (a nucleophile) attacks the core Si atom (an electrophile) from the back, resulting in a penta-coordinate trigonal bi-pyramidal transition state. The nucleophile and Si core then form a new bond, and the leaving group (alcohol) and Si center experience a bond cleavage.

### 2.2. Classification of Silanes

Silanes are basically classified as functional and non-functional silanes based on the presence of active functional groups, as shown in Figure 4. As already detailed, organofunctional silanes will have functional groups such as amino, mercapto and vinyl groups, along with three alkoxy groups. Two separate reactive functional groups are present in organofunctional silanes, which can react and pair with a variety of inorganic and organic compounds. Organofunctional silanes thus serve as adhesion promoters to increase the union of disparate elements. The surface hydroxyl groups of inorganic substrates are reacted with the hydrolyzable functional groups. Organofunctional silanes can react with different functional groups of various compounds [45]. Types of organofunctional silanes and their applications are listed in Table 2. 

On the other hand, non-functional silanes will contain only reactive alkoxy (-OR) functional groups. These groups are hydrolyzed to silanol groups and react with the surface hydroxyl groups of inorganic substrates. A bis-functional silane is also known as a crosslinking or dipodal silane and has two Si atoms with three hydrolyzable alkoxy groups on each [46]. Some examples of functional and non-functional silanes were given to understand the classification based on functional groups present in Figure 5 [47]. 

### 2.3. Factors Affecting Silane Hydrolysis

The molecular structure of the silane, its concentration, pH, temperature, humidity and solvent system are some of the important factors that affect the rate of silane hydrolysis. With increasing alkoxy group size, the hydrolysis rate decreases in the order of pentoxy > butoxy > propoxy > ethoxy > methoxy. pH has a significant impact on silane hydrolysis. The rate of silane hydrolysis is rapid in acidic and alkaline media, although it is at its lowest for alkoxysilanes at neutral pH [48]. 

The hydrolysis reaction rate rises as the temperature rises, following the Arrhenius law. The type of co-solvent in the solvent combination also affects the hydrolysis rate. The hydrophilicity of the solvent affects the rate of hydrolysis. The hydrolysis rate of *α*-silanes and *γ*-silanes reduces when the hydrophilicity of methanol, ethanol and propan-1-ol increases. The molecular structure of the terminal silyl group is the cause of the slow crosslinking kinetics of conventional silane-terminated polymers.

The moisture-induced crosslinking reaction occurs much more slowly with *γ*-alkoxysilanes than it does with the extremely reactive *α*-alkoxysilanes. The electron donor is joined to the Si atom in *α*-silanes by a methylene group. The alkoxy groups are activated in this arrangement, greatly accelerating the crosslinking reaction. Figure 6 illustrates the distinction between *α* and *γ*-silanes. Organofunctional alkoxysilanes contain at least one alkoxy group as one of the four groups linked to the Si atom (–OR). Bi-functional (two alkoxy group) and tri-functional (three alkoxy group) silanes are distinguished based on the number of alkoxy groups. Alkoxy groups have the capacity to hydrolyze. A siloxane network is created during the reaction with water. In addition to the alkoxy groups, there is a functional organic group on the Si atom (R) through which a silane can also bind to an organic molecule. The length of the hydrocarbon chain (spacer) in the reactive organic group is a crucial structural component of organofunctional alkoxysilanes. The length of the hydrocarbon chain has a major influence on how firmly the alkoxy groups are bound to the Si atom and, thus, on the speed of crosslinking in the presence of moisture [45].

## 3. Hyperbranched Polymers

HBPs are a well-known family of polymeric materials and are regarded as highly desirable specialized products. It is confirmed that they create an opportunity for the creation of new products, but they also pose a challenge because of their intricate branching structure. It is shown that the structure of HBPs with all distinct structural units may be thoroughly elucidated, allowing one to assess the degree of branching, confirm side reactions, and take into account the kinetics and mechanism of the reaction. The features of HBPs, such as glass transition temperature, solubility and miscibility, melt viscosity, as well as surface qualities, are considerably determined by the alteration of the end groups and define the majority of conceivable outcomes [49].

The theoretical aspect of branched polymer synthesis was first proposed by Flory in 1952, followed by many successful synthetic strategies to produce HBPs. The synthesis of HBPs can be divided into three main strategies: (a) step-growth polycondensation of AB_x_ and A_2_+B_3_ monomers, (b) self-condensing vinyl polymerization of AB monomers and (c) ring-opening polymerization of latent AB_x_ monomers. Utilizing these polymerization strategies, a wide variety of hyperbranched architectures have been synthesized successfully, including polyesters, polyamides, polycarbonates and polyurethanes. HBPs have been utilized in applications in various fields ranging from additives, crosslinkers, coatings and sensors. 

HBPs have attracted significant interdisciplinary research over the past decade. In an ongoing effort to demonstrate the full potential of HBPs, new synthetic techniques are being used to create ever-complex hyperbranched structures. There are a very large number of uses for HBPs that have been researched. Some of these have already been commercially successful. HBPs may have significant cost-saving potential when used in separation procedures such as extractive distillation, solvent extraction, absorption, membranes or preparative chromatography [50].

Three-dimensional macromolecules known as HBPs have many branches. They have distinctive structures and characteristics, including many functional groups, intramolecular cavities, low viscosity and great solubility thanks to their globular and dendritic architectures. HBPs can be easily produced via one-pot polymerization. The significant advancement in synthetic techniques ranging from click polymerization to the recently reported multi-component reactions has resulted in a variety of HBPs with desirable functional groups. Due to their adaptable architectures and distinctive features, HBPs have found extensive use in numerous fields, including coatings, adhesives, modifiers, biomaterials, supramolecular chemistry, hybrid materials and composites, light-emitting materials, nanoscience and technology and supramolecular chemistry. In recent years, HBPs have garnered increasing amounts of interest supported by the utilization of varying monomers to create HBPs and improved synthesis processes. The requirements for reaction conditions and monomer structure vary between different synthesis techniques, and therefore, the final characteristics of HBPs also vary [51,52,53].

### 3.1. Synthetic Strategies of HBPs

HBPs were mainly synthesized by two methodologies, viz. single monomer methodology (SMM) and double monomer methodology (DMM) [54], as shown in Figure 7.

Step-growth polymerization, condensation polymerization, ring-opening polymerization and free radical polymerization have become the most popular methods for synthesizing HBPs. Because of the widespread use of HBPs, researchers have also attempted to create them using certain unique synthesis techniques, including click-chemistry, cross-coupling reaction, dipole cycloaddition reaction and coupling monomer approach [55,56,57].

#### 3.1.1. Step-Growth Polymerization

This methodology involves the polymerization of AB_x_ (x ≥ 2) monomers via one-step polycondensation. The primary advantage of this approach is that normal step-growth polymerization characteristics are obeyed. However, the main drawbacks include gelation, which often occurs during polymerization. A monomer with the functionality of three or more can form HBPs and can reach a gel point forming a crosslinked network structure. The step-growth polymerization reaction can be simply quenched prior to reaching the gel point. Another drawback is that the AB_x_ monomers employed have to be synthesized prior to polymerization, and this is a distinct disadvantage for commercial applications. However, the step-growth polycondensation process offers a diverse synthesis of HBPs using a variety of available monomers, which provides the potential for the preparation of a wide spectrum of functionalities.

#### 3.1.2. Condensation Polymerization

The process of repeated condensation reactions between monomers with two or more active groups to create polymers while releasing water and other small molecules is known as condensation polymerization. The majority of polycondensation processes are reversible and progressive, and as the reaction time increases, the molecular weight gradually rises. However, the rate of monomer conversion is essentially independent of time. A popular technique for creating HBPs is polycondensation of AB_n_-type monomers. By using this technique, numerous researchers have produced polyphenylene, polyether, polyester and polyamide with considerable success. Initially, AB_2_-type monomers, such as ethanolamine, ethylene glycol amine and aminobenzene, were utilized as monomers. Increasing numbers of monomers are employed as this technique is widely used. Propylene glycol, trimethylol propane and other AB_3_ and AB_4_-type monomers are often used [58].

The following are the essential parameters for this reaction of AB_n_-type monomers: (1) no additional side reactions will occur between the monomer components A and B during the polycondensation reaction, and (2) the activity of each group in part B is the same, and intramolecular cyclization does not take place in this group. A synthesis method of combining B_m**-**_ with AB_n_-type monomers to create AB_n_ + B_m_-type monomers was proposed in order to better regulate the molecular weight of the polymer and the geometric configuration of the molecular structure. By varying the ratios of B_m_ and AB_n_, the approach can adjust the molecular weight of the target polymer. An example is shown in Figure 8, in which Tris(hydroxymethyl)aminomethane and isophorone diisocyanate were used in the condensation polymerization by Hi et al. to create a hydroxyl-terminated HBP [59]. On the other hand, instead of polycondensation, self-condensing vinyl polymerization can also take place through initiation and propagation as shown in Figure 9 [60]. This process involves the use of monomers that have a vinyl group and one initiating moiety to generate HBPs; the activated species may be a radical, cation or even a carbanion.

#### 3.1.3. Ring Opening Polymerization

The process of turning cyclic compound monomers into linear polymers using ring-opening addition is known as ring-opening polymerization. In the polymerization of glycidol, anionic polymerization of glycidol proceeds as an intramolecular reaction involving growing (alcoholate anions) and dormant species (-OH groups) to form an HBP (Figure 10).

The reaction procedure is straightforward, and just one step is needed to create HBP from the epoxy monomer. This reaction is gentle to perform, produces fewer byproducts than polycondensation, and makes it simple to produce high molecular weight polymers. The fact that this reaction’s raw materials must contain epoxy-based monomer limits the availability of the reactive monomer is a major drawback. There are limited sources of monomers since the monomers needed for ring-opening polymerization require a unique ring structure. However, after terminal modification, the produced polymer is amphiphilic because of the unique structure of the monomer. As a result, its scope of applicability has also been increased [61,62,63].

#### 3.1.4. Free Radical Polymerization

The majority of monomers used in free radical polymerization are alkenes with unsaturated double bonds. The double bond in the monomer is broken during the process, and the addition reaction is repeated numerous times between the molecules, connecting numerous monomers to form the macromolecule. Fréchet et al. were the first to report on the free radical polymerization of HBPs. The utilized monomers typically have a reactive group that can start vinyl polymerization as well as a vinyl group. The reactive group may be cationic, anionic or free radical. Reactive groups can start vinyl group development during the reaction and move to form new active sites during the chain growth phase to keep starting vinyl polymerization [64].

In contradiction with linear polymers, the monomer used for this approach must have numerous vinyl groups in order to produce HBPs. Nevertheless, this approach frequently results in gel throughout the reaction phase, which has a relatively low monomer-to-polymerization conversion rate. Moreover, the equipment cannot be used to gauge the D.B. in a polymer. It is also challenging to generate polymers with fixed molecular weights since the molecular weight dispersion of synthetic polymers is large [65,66]. Nowadays, the synthesis of HBPs is frequently carried out using condensation polymerization, ring-opening polymerization and free-radical polymerization. Despite the fact that many academics have explored several novel techniques for the synthesis of HBPs; their use is still restricted by the uniqueness of the reaction conditions and monomer structure. 

#### 3.1.5. Cross-Coupling Reaction

A relatively recent organic coupling process is the Suzuki reaction, also known as the Suzuki coupling reaction or Suzuki-Miyaura reaction. A zero-valent Palladium (Pd) complex is used to catalyze the reaction, which entails the cross-coupling of aryl or alkenyl boric acid (borate ester) with chlorine, bromine, iodinated aromatics, or olefins. The Sonogashira coupling reaction, which uses a Pd catalyst to create a C-C bond between a terminal alkyne and an aryl or vinyl halide, is another common cross-coupling reaction [67].

#### 3.1.6. Huisgen Reaction

The Huisgen reaction, also called the 1,3-dipolar cycloaddition reaction, is an alkene, alkyne, or derivative of an alkene or an alkyne cycloaddition reaction. The end product is a heterocyclic molecule with five members [65]. Ye et al. used azide-alkyne to create an HBP with a poly (-caprolactone) (PCL) chain and a polystyrene (PS) chain based on click chemistry through Huisgen cycloaddition reaction [68].

#### 3.1.7. Coupling Monomer Method

Gao and Yan employed the coupling monomer approach to have the two monomers produce AB_n_-type intermediates in the reaction system (Figure 11). After additional polymerization, they were able to produce HBPs. Currently, this technique has been used to successfully manufacture polyamide, polyethoxylsilane, polyamide and polyester-amide [69].

### 3.2. Degree of Branching in HBPs

The structural properties of HBPs having a highly branched 3D structure with a dendritic-like architecture, place them between conventional linear polymers and dendrimers (Figure 12). The D.B. in these polymers is one of the crucial factors to consider while describing them. A linear polymer’s D.B. is 0, a dendrimer’s is 1, and HBPs fall between these two with a D.B. of 0.4 to 0.6. The features of the HBPs, such as low molecular entanglement, low melting/solution viscosity, high solubility, host-guest contact capability, and self-assembly behavior, would thus be greatly influenced by the D.B. value. Importantly, each of these characteristics is tunable through modifying branches, end-groups and D.B. [70,71,72,73].

The D.B.% in systems involving the direct condensation process between bi-functional A_2_ monomers, such as diacids, and tri-functional B_3_ monomers, such as glycerol, in the presence of lipase and chemical catalysts to generate HBPs, has been thoroughly researched. To find prospective applications, it is important to comprehend the connection between D.B.% and the physical and chemical characteristics of HBPs. The proportion of linear (L), terminal (T), and dendritic (D) units in the matrix of the polymer determines the degree of branching, which is a structural property [74,75,76].

The constituents are dendritic units, fully incorporated with an AB_x_ monomer, terminal units having two unreacted B groups and linear units retaining one unreacted B group. The linear segments are generally described as defects. Fréchet et al. defined the term ‘degree of branching (D.B.)’ as
D.B. = *(D + T)/(D + L + T)*(2)
where, *D*, *T* and *L* are the number of dendritic, terminal and linear units, respectively. D.B. is one of the important characteristics that indicate the branching structure of HBPs [77]. Frey and colleagues reported a modified definition of D.B. based on the growth directions as
D.B. = *2D/(2D + L)* = *(D + T − N)/(D + L + T − N)*(3)

Where, *N* is the number of molecules. The two equations give almost the same D.B. for HBPs with high MWs, as ‘*N*’ in Frey’s equation can be negligible in such cases [71]. In general, it is pointed out that D.B. is determined by statistics and only reaches 50% for the polymers derived from the AB_2_monomer, assuming the equal reactivity of the B functional group towards the A functional group. It should also be noted that HBPs possess many isomers even if D.B. is equal to 100%.

### 3.3. Role of Silanes in HBPs

Silanes play an important role in the synthesis and modification of HBPs. They act as crosslinking agents, coupling agents or end-capping agents to improve the mechanical and thermal properties, solubility, stability and compatibility of HBPs with various substrates. Additionally, functional silanes can introduce specific functionalities into the HBP structure, leading to new applications in various fields such as biomedical engineering, electronics and energy storage. In the synthesis of HBPs, silanes serve as multi-functional building blocks to control the branching structure, molecular weight and functionalities of the resulting polymers. The choice of silanes and their ratios can significantly impact the resulting HBP properties, making it a crucial factor in the synthesis of HBPs. Here are some specific ways silanes contribute:

Cross-linking agents: Silanes with reactive functional groups (e.g., epoxy, amine, or hydroxyl) can act as cross-linking agents to promote the branching structure and enhance the mechanical properties of HBPs.

Coupling agents: Silanes with reactive functional groups (e.g., isocyanate or epoxy) can act as coupling agents to increase the molecular weight of HBPs and improve their solubility in various solvents.

End-capping agents: Silanes with reactive functional groups (e.g., isocyanate or epoxy) can act as end-capping agents to terminate the growing polymer chains and improve the stability of HBPs.

Functionalizing agents: Silanes with specific functional groups (e.g., carboxylic acid, amine, or epoxy) can introduce specific functionalities into the HBP structure, leading to new applications in various fields, such as biomedical engineering, electronics and energy storage.

### 3.4. HBPs Containing Si Atom

Zhang et al. successfully prepared a series of novel hyperbranched polycarbosiloxanes by the Piers–Rubinsztajn (P–R) reactions of methyl-, or phenyl-triethoxylsilane and three Si–H containing aromatic monomers, including 1,4-bis(dimethylsilyl) benzene,4,40-bis(dimethylsilyl)-1,10-biphenyl and 1,10-bis(dimethylsilyl)ferrocene using B(C_6_F_5_)_3_ as catalyst for 0.5 h at room temperature. Their structures were fully characterized by FTIR, ^1^H-, ^13^C- and ^29^Si-NMR. The molecular weights were determined by gel permeation chromatography. The D.B. of these polymers was 0.69–0.89, which was calculated based on the quantitative ^29^Si-NMR spectroscopy [78].

Liu et al. synthesized many hyperbranched tetrahedral polymers by facile Suzuki coupling polycondensation reactions between tetrabromoarylmethane silane and 9,9-dihexylfluorene-2,7-diboronic acid at low concentrations. These polymers were soluble in common organic solvents such as THF and DMF and exhibited excellent thermal stability. The polymers exhibited strong blue fluorescence under excitation by UV light in the solution and the solid state. The polymers were less prone to self-aggregation in the solid state due to their hyperbranched structures, and no excimer-like long wavelength emissions were observed in their solution and film PL spectra [79].

Hartmann-Thompson et al. attempted to produce a series of novel hyperbranched hydrogen-bond acidic polymers were prepared by functionalizing hyperbranched polycarbosiloxanes or polycarbosilanes with phenol or hexafluoro-2-propanol groups. Starting polymer, sensor polymer and reagent structures were confirmed by IR, ^1^H-, ^13^C- and ^29^Si-NMR, SEC or GCMS as appropriate [80].

A controllable approach to synthesize UV curable hyperbranched polysiloxysilanes from A_2_ (1,1,3,3-tetramethyldisiloxane) and CB_2_ type monomers (methyl(vinyl)silanediylbis(oxy)bis(ethane-2,1-diyl) diacrylate and methyl(vinyl) silanediylbis(oxy)bis(ethane-2,1-diyl)bis(2-methylacrylate)) was developed by Metroke et al. Polymerization was monitored using FTIR, where a two-step poly-addition mode was observed. The vinyl silane group preferentially reacted with hydride silane, resulting in the formation of AB_2_ type intermediates containing one hydride silane and two acrylate (or methacrylate) groups, and at the same time, there may be a low quantity of B_4_ type intermediates. The intermediates further polymerized to form HBPs, and their kinetics was also studied [81].

Chen et al. successfully developed a novel hyperbranched poly(phosphamide) containing abundant silanes as terminal groups (HBPPA-Si) synthesized by the A_2_ + B_3_ polymerization reaction, which can be used as a charring and blowing agent simultaneously. HBPPA-Si exhibited good performance of char formation and high thermal stability. The char residue at 800 °C was 44 wt. -% under a nitrogen atmosphere [67].

Zhu et al. successfully prepared a novel hyperbranched organosilicon polymer via step-growth thiolene click reaction using mercaptopropyl methyl diallyl silane (AB_2_) and mercaptopropyltriallylsilane (AB_3_) as hyperbranched monomers. The structures of the prepared polymers were characterized by FTIR and NMR spectroscopy. The D.B. of the polymers was determined using quantitative ^29^Si-NMR spectroscopy, and the D.B. of the polymers from AB_2_ and AB_3_ was 0.60 and 0.22, respectively [82].

Xue et al. prepared three hyperbranched polysiloxanes with methacrylate (HSiM), epoxy (HSiG) and methacrylate/epoxy groups (HSiMG) through the hydrolysis of γ-glycidoxypropyltrimethoxysilane (γ-GPS), γ-methacryloxypropyltrimethoxysilane (γ-MPS) and their binary blends. The results indicated that methacrylate groups could accelerate the condensation reaction, thus producing more branching sites with a branching degree of 0.67 for HSiMG. In addition, the effects of terminal groups on the microstructural and thermal properties of the post-cured hyperbranched polysiloxanes were evaluated [83]. Silane HBPs were also used as core material, toughening agent, optical emission inducer, surface coating agent, coupling agent and surface modifier as shown in Table 3 [84,85,86,87,88,89].

## 4. Silane Polymers Applications

### 4.1. Silanes in Surface Coatings

Commercially, silanes and organically modified silanes are utilized in indoor applications to reduce water vapor condensation and microbial contamination as well as to preserve the built environment from deterioration. Ciriminna et al. highlighted the importance of silane coatings in various fields. Monomeric silane molecules can pass through 1nm pores and chemically bond with building materials through the OH groups at the material’s outer surface before polymerizing within the pores to effectively block water from entering the building envelope (Figure 13). Silane-based paints, particularly those in the form of silicates and organically modified silica, are successfully used to protect the built environment, including modern and historic buildings as well as marine structures, due to their exceptional versatility, long lifespan and unique ability to maintain the aesthetics of the treated surfaces. Silane paints coat the internal porosity of a substrate with hydrophobic layers using liquid monomeric alkylsilanes [91].

Organosilica-based nanosols often cling to substrate surfaces via the formation of siloxane linkages with hydroxyl groups at the substrate surface. The surfaces of materials such as glass, fiberglass, polycarbonate, wood, cotton, stone, concrete, marble, iron and aluminum (where numerous OH groups are present due to partial metal oxidation) are consequently coated easily and then require only simple curing at room temperature. Hence, anti-corrosion and water-resistant sol-gel organically modified silica coatings are well suited to be applied to the surface of the majority of building surfaces.

### 4.2. Silanes in Corrosion Resistance

Silanes are used in corrosion resistance of metals based on their ability to form essential silanol groups (-SiOH) through hydrolysis followed by condensation, which allows bonding with the surface functional groups (MeOH) of metals and silanol groups as well as among silanol groups themselves, as shown in Figure 14. Following the condensation (i.e., curing or drying process) of -SiOH and MeOH groups with release of water, a siloxane network (Si-O-Si) is developed as a result of the reaction between silanols and metallo-siloxane linkages (Me-O-Si) **[92]**.

If a carbon atom serves as one of the Si atom’s substituents, the as-formed Si-O-Si network becomes extremely hydrophobic. The creation of the siloxane network, or Si-O-Si, is essential for corrosion protection due to its hydrophobic properties. The method of bonding between silane molecules and the metal’s surface hydroxide layer is depicted schematically [93,94].

### 4.3. Silanes as Additives

The hydrophobic properties of silane allowed the silane additives to significantly increase the mortar’s imperviousness and resistance to freeze-thaw. Due to the silane’s bridging property, its mortar’s crack resistance was increased. The inherent benefits of siloxane, such as carbonation and chemical resistance, improved the mortar’s resistance to carbonation. With mortars containing 1% silanes, the silanes tightly enclose the mortar particles. The network of silane polymers becomes denser as the silane dosage is increased further. These results contribute to the explanation of the increase in mortar durability following the addition of silane. Silanes have a networking effect that ties mortar particles together, improving the mortar’s resistance to cracking, carbonation corrosion and freeze-thaw behavior [95].

### 4.4. Silane Polymers as CrossLinking Agents

Silane polymers can be used as crosslinking agents in various industries, including construction, coatings, adhesives and rubber [96]. In construction, silane polymers can be used as a crosslinking agent to improve the strength, durability and resistance to freeze-thaw cycles of concrete and cement. When added to the concrete or cement mix, silane polymers can crosslink with the cement hydrate and aggregate particles, strengthening the bonds between them and increasing the compressive strength of the concrete. In coatings, silane polymers can be used as a crosslinking agent to improve the durability and resistance of the coating to weathering, UV radiation and chemical attack. They can also be used to improve the adhesion of the coating to the substrate. In adhesives, silane polymers can be utilized as a crosslinking agent to improve the strength and durability of the adhesive bond. They can also be used to improve the resistance of the adhesive to temperature, humidity and chemical attack. In rubber, silane polymers can be used as a vulcanizing agent to improve the strength and durability of the rubber. They can also be used to improve the resistance of the rubber to heat and ozone, as well as its flexibility. In general, crosslinking of silane polymers can improve the mechanical and thermal properties of the materials, making them more durable and resistant to environmental factors [97].

### 4.5. Silane Polymers in Sensor Applications

Chen et al. developed a unique technique that involves the easy, one-step targeted immobilization of an enzyme without the use of complicated purification procedures. For this purpose, the silane emulsion self-assembly process is carried out using horseradish peroxidase as a template, 3-aminopropyltriethoxysilane (APTES) is used as a functional monomer, and tetraethyl orthosilicate (TEOS) is used as the crosslinking agent to form a unique molecularly imprinted polymer. Visual sensors for the detection of glucose and sarcosine were created using the immobilized horseradish peroxidase. This study proved that horseradish peroxidase may be easily immobilized from a horseradish crude extract using molecularly imprinted polymers created using the silane emulsion self-assembly approach without the need for any purification steps and shows great potential applications for the visual detection of glucose and sarcosine [98].

Trovato et al. produced a halochromic sensing molecule using Alizarin Red S with trimethoxy-[3-(oxiran-2-ylmethoxy) propyl] silane (GPTMS) as shown in Figure 15 to encapsulate polyester fabrics. The outcome was a pH-responsive color change in the polyester fabrics treated with the optically transparent organic-inorganic halochromic coatings in contrast to plain ARS treated polyester fabrics, which totally leach the halochromic molecules after the first washing cycle. The polyester fabrics coated with silane functionalized Alizarin Red showed a considerable dynamic reactivity to pH variations even after many washing cycles. The cross-linkage between the alkoxysilane network and the surface functional groups of polyester textiles is further supported by coating adhesion studies carried out by gravimetric and spectrophotometric calculations [99].

Herlem et al. modified gold surfaces with APTES and an electrolyte based on THF solvent to make thin films. These films were identified using X-ray photoelectrons and IR-ATR spectroscopies. By using scanning tunneling microscopy, the film shape was examined, and ellipsometry measurements were used to track the film’s growth. The results obtained indicate that the chemical composition of the electrodeposited films in liquid APTES and liquid THF is the same. This amino-terminated coated film has major significance for sensing applications, which was used as the functionalized component of a surface plasmon resonance biosensor to monitor lactalbumin graft [100].

Kros et al. prepared various hybrid silane materials and their potential applications in the biomedical field. The creation of biocompatible coatings is based on sol-gel silicates, which can be utilized as a protective covering for implantable glucose sensors. The characteristics of the resulting sol-gel composites were modified when silica was blended with various organic polymers. Their uses on biosensors and their biocompatibility invivo and in vitro were examined. An amphiphilic block copolymer is made up of segments of hydrophobic poly (methylphenylsilane) and blocks of hydrophilic poly (ethylene oxide). This polymer creates vesicles in an aqueous media that contain a fluorescent dye, which can be utilized in drug delivery systems [101]. The versatility and unique properties of silane polymers make them a promising material for use in a variety of sensors, including those for medical, environmental, and industrial applications. Silane polymer-based sensors are used in the medical field for various applications, such as continuous monitoring of glucose levels in diabetics, Heart rate and body temperature monitoring, detecting the presence of specific biomolecules (e.g., proteins), drug delivery systems and tissue engineering for regenerative medicine.

## 5. Conclusions

Organosilicon polymers are a class of highly important inorganic-organic compounds with high molecular weights. In other words, Si-containing HBPs are compounds of high technological importance. Many potential applications exist for this class of compounds due to its multi-functional properties, such as surface coatings, corrosion resistance, coupling agents, additives, crosslinkers and sensors. This article reviews the important synthetic methods of silane-based HBPs and the basic properties of silanes that have been developed over the past decade. As each and every individual polymer have different parameters, it is clear that selecting the appropriate silane is essential when developing high-performance HBPs. The fact that materials containing Si may be produced on an industrial scale makes this point especially important. In addition, silane-based HBPs can be employed in various fields, especially as sensors and biosensors for the detection of both single molecules and a large group of substances. Si-HBPs serve as an ideal matrix for anchoring biological material since they not only successfully attach to its structure but also enhance the performance characteristics of bio-recognition components. Due to its favorable mechanical, thermal and conductive characteristics, frameworks on silane-HBPs are grabbing the focus of scientific teams more and more as they can be used well in a wide range of applications in multiple fields.

## Figures and Tables

**Figure 1 polymers-15-02517-f001:**
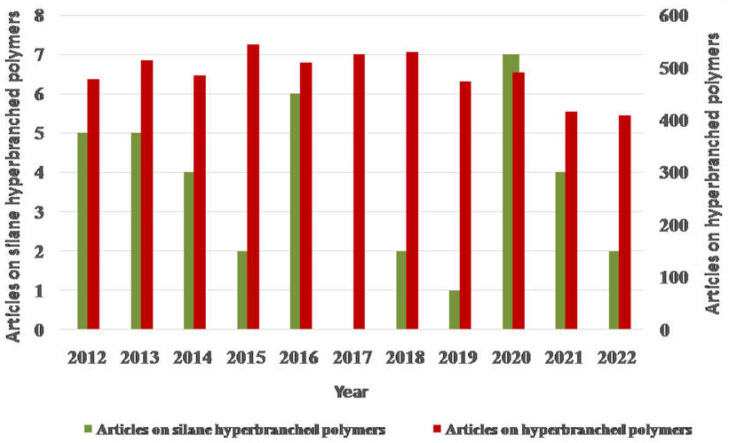
Statistics of literature retrieved from the Web of Science portal with the theme of “HBPs” and “Si-based HBPs”.

**Figure 2 polymers-15-02517-f002:**
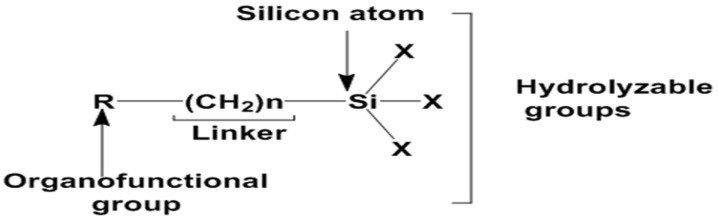
The chemical structure of organofunctional silanes with the Si atom, hydrolyzable alloy groups, linker and organofunctional group [38].

**Figure 3 polymers-15-02517-f003:**
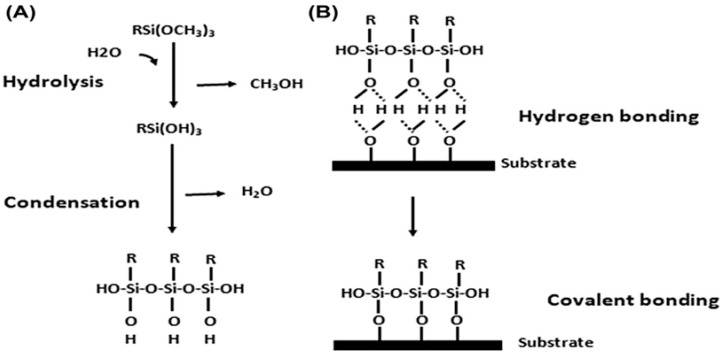
Hydrolysis and condensation mechanism of organofunctional silanes and bonding mechanism to a substrate. (**A**) Hydrolysis and condensation to form polymers, and (**B**) adsorption to an inorganic substrate (such as ceramics or surface oxide layers on metals) by hydrogen bonding and then covalent bonding to the substrate by a condensation reaction with hydroxyl groups. Reproduced from [44].

**Figure 4 polymers-15-02517-f004:**
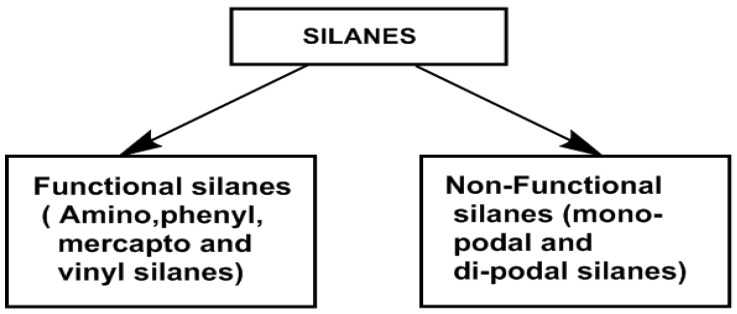
Classification of silanes as functional and non-functional silanes. Adapted from [45].

**Figure 5 polymers-15-02517-f005:**
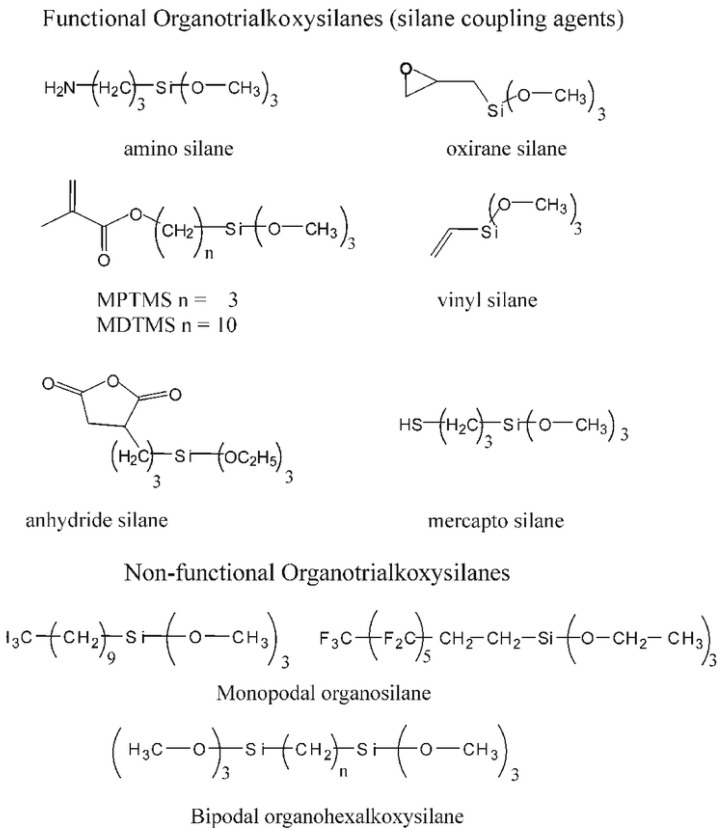
Structural examples for functional and non-functional silanes. Reproduced from [47].

**Figure 6 polymers-15-02517-f006:**
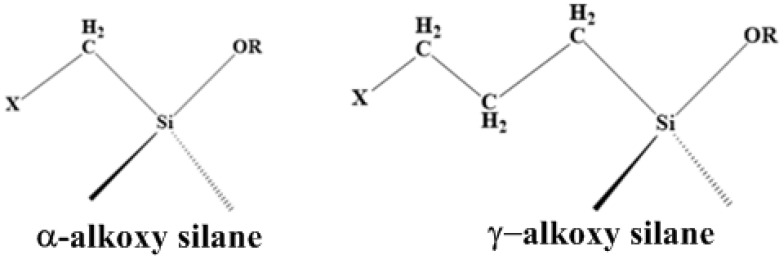
Difference between *α*-alkoxy silanes and *γ*-alkoxy silanes configuration. Adapted from [45].

**Figure 7 polymers-15-02517-f007:**
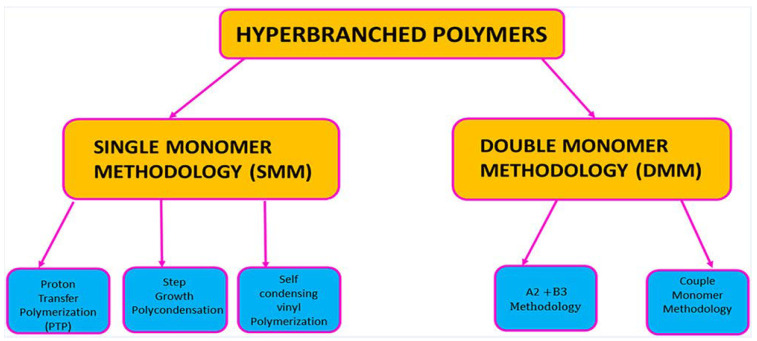
Schematic illustration of various synthetic strategies used to prepare HBPs. Reprinted with permission from [18] Copyright 2023, American Chemical Society, Washington, DC, USA.

**Figure 8 polymers-15-02517-f008:**
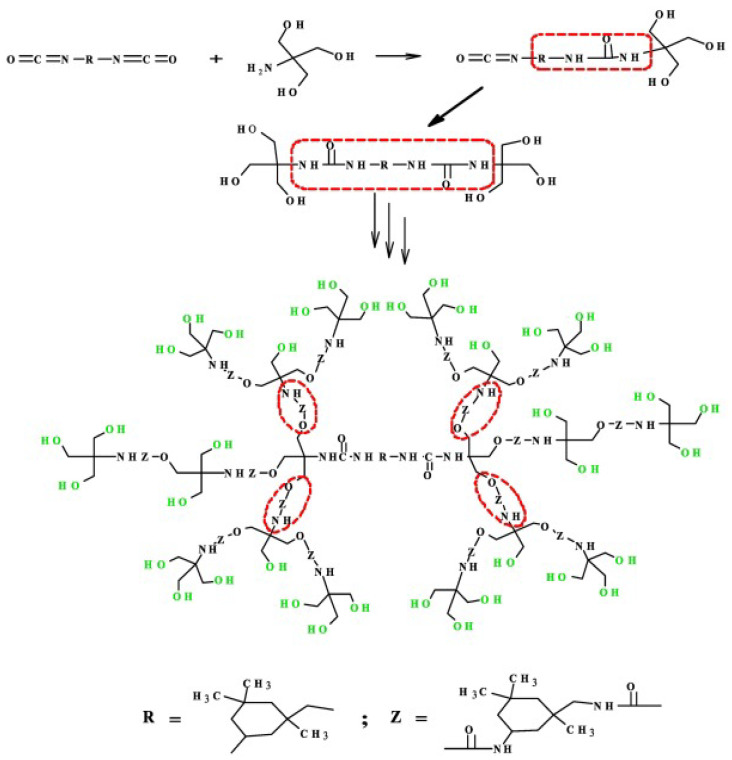
Schematic illustration of condensation polymerization. Reprinted with permission [59] from Elsevier.

**Figure 9 polymers-15-02517-f009:**
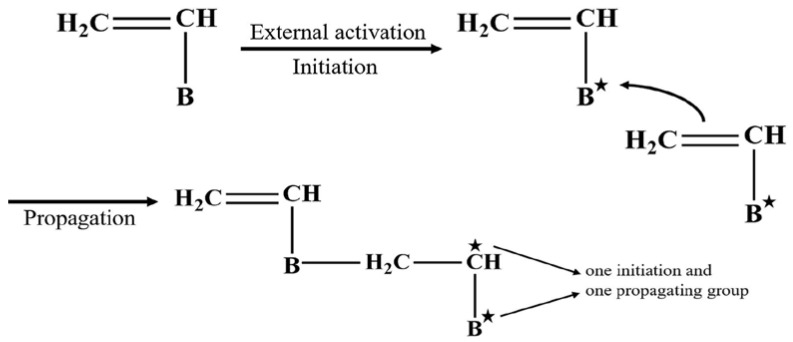
Self-condensing vinyl polymerization. Reprinted with permission [60] from Elsevier.

**Figure 10 polymers-15-02517-f010:**
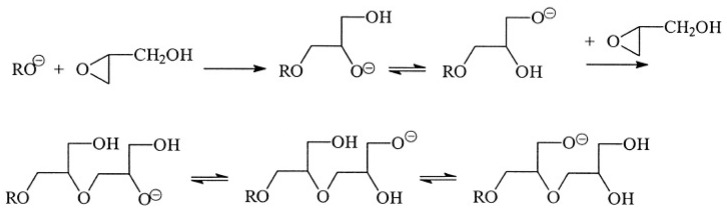
Schematic representation of ring-opening polymerization mechanism. Reproduced with permission [63]. Copyright 2023 John Wiley and Sons.

**Figure 11 polymers-15-02517-f011:**
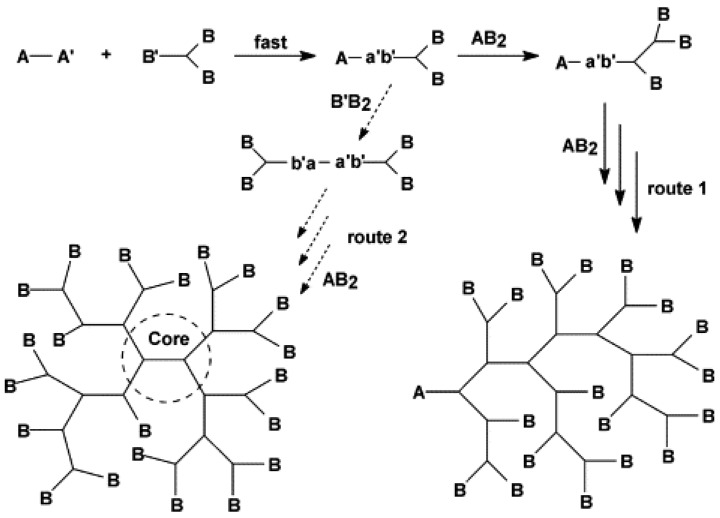
Schematic representation of ring-opening polymerization mechanism. Reproduced with permission [69] from Elsevier.

**Figure 12 polymers-15-02517-f012:**
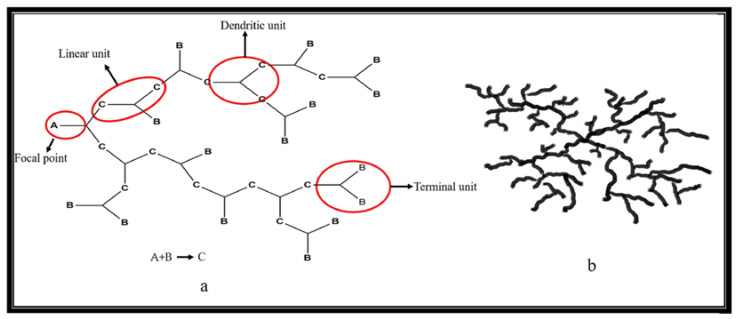
(**a**) Degree of branching in HBPs. (**b**) Schematic representation of HBP. Reprinted with permission [60] from Elsevier.

**Figure 13 polymers-15-02517-f013:**
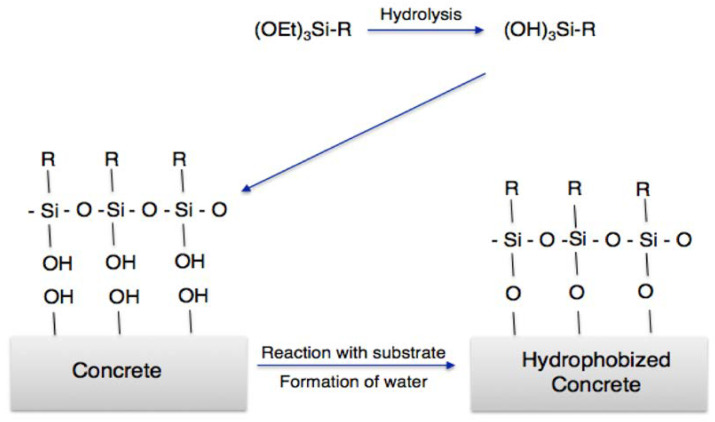
Bonding of silanes with building surface through hydrolysis. Reproduced with permission [91].

**Figure 14 polymers-15-02517-f014:**
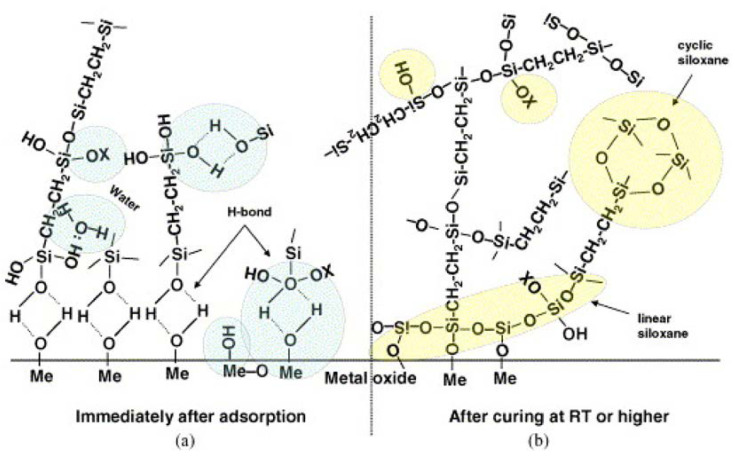
Schematic bonding mechanism between silanes and metal surface: (**a**) before condensation, and (**b**) after condensation [92].

**Figure 15 polymers-15-02517-f015:**
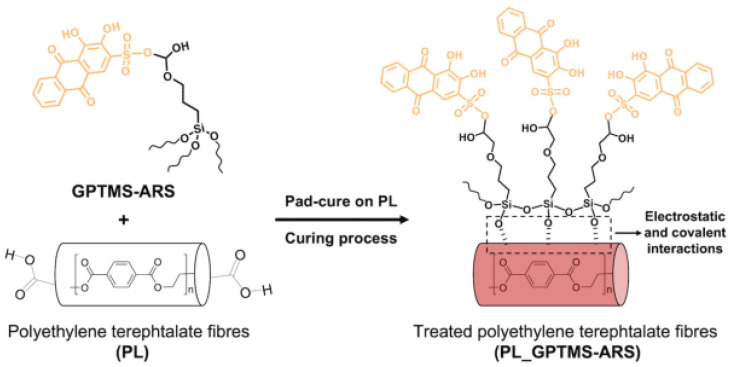
Schematic representation of GPTMS-ARS pad cure process on PL fibers. Reproduced from [99].

**Table 1 polymers-15-02517-t001:** Examples of organofunctional groups with reactive functional groups.

S. No	Name of the Silane	Structure	Organofunctional Group Present
1.	Trichloromethoxy silane	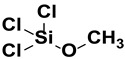	Methoxy group
2.	Trimethoxyphenyl silane	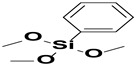	Phenyl group
3.	ϒ-mercaptopropyl trimethoxy silane	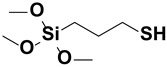	Sulfo group
4.	ϒ-aminopropyl trimethoxy silane	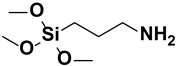	Amino group
5.	ϒ- glycidoxypropyl-trimethoxysilane	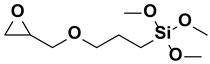	Epoxy group
6.	Trichlorovinyl silane	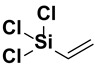	Vinyl group
7.	Triethoxyvinyl silane	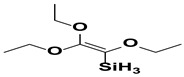	Vinyl group

**Table 2 polymers-15-02517-t002:** Types of silane and their applications.

Type of Silane	Applications
Alkoxy/chloro silanes	Blocking agent, surface modification and coatings, coupling agent
Amino silanes	Coupling agent, adhesion promoter, glass fiber reinforcement, cross-linker, pigment dispersion
Phenyl silanes	Coupling agent, industrial coatings, surfactants, hybrid materials
Mercapto silanes	Fillers, composites, coupling agents, adhesion promoters
Vinyl silanes	Coupling agent, adhesion promoters, crosslinkers

**Table 3 polymers-15-02517-t003:** Silane HBPs and their properties and applications.

S. No	Type of Silane-HBP	Properties	Applications	Ref.
1.	Ferrocene linked Si-HBP	Precursors in ceramics construction,	High technology applications	[78]
2.	Hyperbranched tetrahedral polymers	Electro-luminescence emitters	Photoluminescence Light emitting diode	[79]
3.	Polycarbosiloxanes polymers	Surface acoustic wave generator	Sensor applications	[80]
4.	polysiloxysilanes	Photo-initiators	Photo-initiating process	[66]
5.	Poly phosphamide silanes	Composites for charring and blowing	Flame retardants	[81]
6.	Organosilicon polymers	Crosslinkers	Heavy metal absorption	[82]
7.	Hyperbranched polysiloxanes	Microstructures with more branching sites	Structural designing	[83]
8.	Silane-modified alkyd polymer	Core material for polyurethane coating	Eco-friendly coatings	[84]
9.	Hyperbranched Poly (diethynyl benzene Silane)	Optical properties	Optical emission	[85]
10.	Silane functionalized graphene oxide-HBP	Stabilizer	Toughening agent	[86]
11.	Amine terminated HBPI-silica hybrid	Coupling agent	Surface modification	[87]
12.	Hyperbranched Polysiloxane	Surface wettability and UV resistance	High-performance fibers	[88]
13.	Hyperbranched polyurethane-urea-imide/o- silica hybrids	Surface modifier	Hybrid coatings	[89,90]

## Data Availability

Not applicable.

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
