# Peer review of "A Comprehensive Review on Processing, Development and Applications of Organofunctional Silanes and Silane-Based Hyperbranched Polymers"

_polymers, 2023, doi:10.3390/polym15112517_

Round 1

Reviewer 1 Report

In this review titled as “A Comprehensive Review on Processing, Development and Applications of Silane based Hyperbranched Polymers”, the authors claimed that this review focuses on the research progress in silane based HBPs and their applications in the last decade. However, after careful evaluation, I really do not think it can be published at present due to the poor quality. Following are some detailed comments.

1.       The resolution of all figures is really poor, and has not reach the requirement for publication. Please redraw them. In addition, some figures, such as Figures 14 and 18 are incomplete. I clearly remembered some figures, Figures 11 and 12, have been published. But the authors have not cited any reference. This may be suspected of plagiarism.

2.       Some data are obviously wrong. Taken Figure 1 as an example, I have checked in “web of science”, the number of published papers on HPBs are much more than the authors claimed.

3.       The theme of this review is in fact not clear. From the title and abstract, this review should focus on Silane based Hyperbranched Polymers. However, this thing was firstly appeared on Page 14 (this review is only 21 pages without references). This is obviously unjustified.

4.       The citation is unfair at all. For example, in Page 14, they said “Fréchet et al. defined the term ‘degree of branching (DB)’ as….,” and “The two equations give almost the same DBs for HBPs with high MWs as N in Frey's equation can be negligible in such cases”. However, in this paragraph, none of their work has been cited. Instead, they cited the following paper “Murillo, E.A.; Vallejo, P.P.; Sierra, L.; López, B.L. Characterization of Hyperbranched Polyol Polyesters Based on 2,2-Bis 807 (Methylol Propionic Acid) and Pentaerythritol. J. Appl. Polym. Sci. 2009, 112, 200–207, doi:10.1002/app.29397”. The cases like that are really terrible, and may also be suspected of academic misconduct.

5.       Please be noted that in a chemical formula, the number should be subscript.

6.       The language is also really poor.

Author Response

Response to Reviewer-1 Comments

Point 1: Reviewer recommend to improve the resolution of all figures and add citation to figures.The resolution of all figures is really poor, and has not reach the requirement for publication. Please redraw them. In addition, some figures such as Figures 14 and 18 are incomplete. I clearly remembered some figures, Figures 11 and 12, have been published. But the authors have not cited any reference. This may be suspected of plagiarism.

Response 1: We agree with the reviewer’s comment. To support this explanation, Figures 1,2 and 4 were redrawn. Citation given to all the figures including copyright permission for relevant figures (incl. Figs. 11, 12 and 14). Figure 18 removed due to copyright issue.

Point 2: Reviewer recommend to recheck the data given in Fig. 1.Some data are obviously wrong. Take Fig. 1 as an example, I have checked in “Web of Science”, the number of published papers on HBPs are much more than the authors claimed.

Response 2: We agree with the reviewer’s point. To support this explanation,we attached the downloaded data from Web of Science.In total, 5,391 articles were published on the topic “hyperbranched polymers” in the past decade (2012-2022). Earlier, we mentioned that there are more than 400 papers published per year in the past decade. We changed that into “above 500 articles were published per year” in the revised version.

Point 3: Reviewer commented that the theme of this review is in fact not clear. From the title and abstract, this review should focus on “Silane based Hyperbranched Polymers”. However, this thing was firstly appeared on Page 14 (this review is only 21 pages without references). This is obviously unjustified.

Response 3: A detailed overview on silanes are given in the revised manuscript in order to provide the readers the basic understanding of silanes. Hence, the title is also renamed as “A Comprehensive Review on Processing, Development and Applications of Organofunctional Silanes and Silane based Hyperbranched Polymers”.

Point 4: Reviewer recommend the citation is unfair at all. For example, in Page 14, they said “Fréchet et al. defined the term ‘degree of branching (DB)’ as….,” and “The two equations give almost the same DBs for HBPs with high MWs as N in Frey's equation can be negligible in such cases”. However, in this paragraph, none of their work has been cited. Instead, they cited the following paper “Murillo, E.A.; Vallejo, P.P.; Sierra, L.; López, B.L. Characterization of Hyperbranched Polyol Polyesters Based on 2,2-Bis 807 (Methylol Propionic Acid) and Pentaerythritol. J. Appl. Polym. Sci. 2009, 112, 200–207, doi:10.1002/app.29397”. The cases like that are really terrible, and may also be suspected of academic misconduct.

Response 4: The citation for Frechet et al. and Frey were given in the beginning of the paragraph “Degree of branching” as Refs. 71, 76 and 77 (in Pages 14 and 15).

Ref. 71: Halter, D.; Frey, H. Degree of Branching in Hyperbranched Polymers. 2. Enhancement of the DB: Scope and Limitations. Acta Polym.1997, 48, 298–309, doi:10.1002/actp.1997.010480802.

Ref. 76: Frey, H.; Hölter, D. Degree of Branching in Hyperbranched Polymers. 3. Copolymerization of ABm-Monomers with AB and ABn-Monomers. Acta Polym.1999, 50, 67–76, doi:10.1002/(SICI)1521-4044(19990201)50:2/3<67::AID-APOL67>3.0.CO;2-W.

Ref. 77. Hawker, C.J.; Lee, R.; Frechet, J.M.J. One-Step Synthesis of Hyperbranched Dendritic Polyesters. J. Am. Chem. Soc.1991, 113, 4583–4588, doi:10.1021/ja00012a030.

Point 5: Please be noted that in a chemical formula, the number should be subscript.

Response 5: All the chemical formulas were re-checked and corrected.

Point 6: The language is also really poor.

Response 6: We have carefully checked and revised the entire manuscript and also corrected all the typo-errors present.

Overall, the authors thank the reviewer for their valuable comments raised after reviewing this paper. We believe that the comments raised by the reviewer have really helped us to improve the quality of this work. We have sincerely tried to reply for the queries raised by the reviewer and hope that the revised manuscript will not only satisfy the reviewer but also be suitable for publication in Polymers (MDPI).

Thank you

Reviewer 2 Report

In the current review work, the authors have systematically introduced the important synthetic methods of silane based HBPs and the basic properties of silanes that have been developed over the past decade. Considering the fact that only countable works regarding the silane HBPs have been published within the past decades, the authors did a good job on covering the related publications. However, as a review work, attentions also need to be paid on the theoretical aspects on the developments of HBPs. Overall, I think the work is useful, but some concerns have to be addressed before I can recommend the publication of the work:

Majors:

1.     According to the authors’ statement: “hyperbranched polymers (HBPs) have gained wider theoretical interest and practical applications in …”, however, none of the theoretical works has been introduced in the current version. The authors need to draw the readers’ attention towards the theoretical developments on HBPs as well.

2.     According to the authors statement, within the last decade, only 19 papers related to silane HBPS have been reported, comparing to the large amount of generic HBPs, which has more than 4000 papers, could the authors explain further what are the significances and advantages of silane HBPs (except for the wider space)?

3.     All figure cations need to be improved, the current form for figure captions is extremely simple. Please make sure that the readers could understand the figure by just reading the figure caption. Also, the copyright statement needs to be added for the cited paper in figure cation as well.

Minors:

1.     L93P2, “improvementof” should write “improvement of”.

2.     Equation 3 has format problem.

Author Response

Response to Reviewer-2 Comments

Point 1: According to the authors’ statement: “hyperbranched polymers (HBPs) have gained wider theoretical interest and practical applications in …”, however, none of the theoretical works has been introduced in the current version. The authors need to draw the readers’ attention towards the theoretical developments on HBPs as well.

Response: We agree with the reviewer’s point. As the main theme of this manuscript is overview on silanes and synthesis of silane based HBPs, theoretical developments of HBPs were not discussed in detail, but a new paragraph is included in Page 9 in the revised manuscript as per the reviewer’s suggestion.

“The theoretical aspect of branched polymer synthesis was first proposed by Flory in 1952 followed by many successful synthetic strategies to produce HBPs. The synthesis of hyperbranched polymers can be divided into three main strategies: (a). step-growthpolycondensation of ABx and A2+B3 monomers, (b). self-condensing vinyl polymerization of AB monomers and (c). ring-opening polymerization of latent ABx monomers. Utilizing these polymerization strategies, a wide variety of hyperbranched architectures have been synthesized successfully, including polyesters, polyamides, polycarbonates and polyurethanes. Hyperbranched polymers have been utilized in applications in various fields ranging from additives, crosslinkers, coatings, nano devices such as sensors”.

Point 2: According to the authors statement, within the last decade, only 19 papers related to silane HBPs have been reported, comparing to the large amount of generic HBPs, which has more than 4000 papers, could the authors explain further what are the significance and advantages of silane HBPs (except for the wider space)?

Response 2: We agree with the reviewer’s point. The organo functional silanes can fulfill the minimum requirement to produce a hyperbranched polymer by readily reacting with the monomer due to its hydrolysable nature with most of the organic/polymeric materials. Development of such silane based HBPs are required to address the complications using conventional core molecules. Silane based HBPs can be effectively used as a surface treatment material and as an excellent adhesive, and it can also used directly in the polymeric material due to its ability to bond within the polymer matrix.The revised discussion is included in Page 2 of the revised manuscript.

Point 3: All figure citations need to be improved, the current form for figure captions is extremely simple. Please make sure that the readers could understand the figure by just reading the figure caption. Also, the copyright statement needs to be added for the cited paper in figure cation as well.

Response 3: All the figure captions are updated and copyright statement is added for the relevant figures in the revised manuscript.

Minor Comments

Point 1: L93P2, “improvementof” should write “improvement of”.

Response 1: The change is made on L93P2 as “improvement of” in the revised manuscript.

Point 2: Equation 3 has format problem.

Response 2: Equation 3 is formatted in the revised manuscript.

DB = 2D/(2D + L) = (D + T − N)/(D + L + T − N)

Overall, the authors thank the reviewer for their valuable comments raised after reviewing this paper. We believe that the comments raised by the reviewer have really helped us to improve the quality of this work. We have sincerely tried to reply for the queries raised by the reviewer and hope that the revised manuscript will not only satisfy the reviewer but also be suitable for publication in Polymers (MDPI).

Thank you

Round 2

Reviewer 1 Report

My suggestion is still rejection.

Reviewer 2 Report

All my original concerns have been addressed properly, thus, I agree the publication of the current version.